# Identification of Lifestyle Risk Factors in Adolescence Influencing Cardiovascular Health in Young Adults: The BELINDA Study

**DOI:** 10.3390/nu14102089

**Published:** 2022-05-17

**Authors:** Jules Morcel, Laurent Béghin, Nathalie Michels, Jérémy Vanhelst, Julien Labreuche, Elodie Drumez, Angela Polito, Marika Ferrari, Laura Censi, Dominique Deplanque, María Luisa Miguel-Berges, Thaïs De Ruyter, Stefaan De Henauw, Luis A. Moreno, Frédéric Gottrand

**Affiliations:** 1INFINITE—Institute for Translational Research in Inflammation (U-1286), Université de Lille, CHU Lille, Inserm, 59000 Lille, France; laurent.beghin@chu-lille.fr (L.B.); jeremy.vanhelst@chu-lille.fr (J.V.); frederic.gottrand@chu-lille.fr (F.G.); 2Clinical Investigation Center, Université de Lille, CHU Lille, Inserm, 59000 Lille, France; dominique.deplanque@univ-lille.fr; 3Department of Public Health and Primary Care, Faculty of Medicine and Health Sciences, Ghent University, 9000 Ghent, Belgium; nathalie.michels@ugent.be (N.M.); thais.deruyter@ugent.be (T.D.R.); stefaan.dehenauw@ugent.be (S.D.H.); 4METRICS—Évaluation des Technologies de Santé et des Pratiques Médicales et Département de Biostatistiques, Université de Lille, CHU Lille, 59000 Lille, France; julien.labreuche@chu-lille.fr (J.L.); elodie.drumez@chu-lille.fr (E.D.); 5Council for Agricultural Research and Economics—Research Center on Food and Nutrition, 00178 Rome, Italy; angela.polito@crea.gov.it (A.P.); marika.ferrari@crea.gov.it (M.F.); laura.censi@crea.gov.it (L.C.); 6GENUD (Growth, Exercise, Nutrition and Development) Research Group, Instituto de Investigación Agroalimentaria de Aragón (IA2), Universidad de Zaragoza, 50009 Zaragoza, Spain; mlmiguel@unizar.es (M.L.M.-B.); lmoreno@unizar.es (L.A.M.); 7Centro de Investigación Biomédica en Red de Fisiopatología de la Obesidad y Nutrición (CIBEROBN), Instituto de Salud Carlos III, 28029 Madrid, Spain

**Keywords:** adolescence, young adults, cardiovascular risk factors, BELINDA study, nutrition, lifestyle

## Abstract

Cardiovascular diseases are the leading cause of mortality worldwide. These diseases originate in childhood, and a better understanding of their early determinants and risk factors would allow better prevention. The BELINDA (BEtter LIfe by Nutrition During Adulthood) study is a 10–14-year follow-up of the HEalthy Lifestyle in Europe by Nutrition in Adolescence study (the HELENA study, a European cross-sectional study in adolescents). The study aims to evaluate cardiovascular risk using the PDAY (Pathobiological Determinants of Atherosclerosis in Youth) risk score during young adulthood (21–32 years), and to examine the impact of risk factors identified during adolescence (12.5–17.5 years). Our secondary objective is to compare the characteristics of the BELINDA study population with the HELENA population not participating in the follow-up study. The HELENA study recruited 3528 adolescents during 2006–2007 and reassessed 232 of them 10–14 years later as young adults. We assessed clinical status, anthropometry, nutrition, physical activity (including sedentary behavior), physical fitness, and mental health parameters, and collected biological samples (blood, stool, and hair). Dietary intake, and physical activity and fitness data were also collected. A multivariable linear regression model will be used for the analysis of the primary outcome. A Chi-square and T-test were conducted for the comparison of the descriptive data (gender, age, weight, height, body mass index (BMI), and maternal school level) between participating and non-participating BELINDA adolescents. When comparing the 1327 eligible subjects with the 232 included in the BELINDA study, no significant differences regarding gender (*p* = 0.72), age (*p* = 0.60), height (*p* = 0.11), and weight (*p* = 0.083) at adolescence were found. However, the participating population had a lower BMI (20.4 ± 3.1 kg/m^2^ versus 21.2 ± 3.6 kg/m^2^; *p* < 0.001) and a higher maternal educational level (46.8% high school or university level versus 38.6%; *p* = 0.027) than the HELENA population who did not participate in the BELINDA study. The complete phenotyping obtained at adolescence through the HELENA study is a unique opportunity to identify adolescent risk factors for cardiovascular diseases. This paper will serve as a methodological basis for future analysis of this study.

## 1. Introduction

Cardiovascular diseases (CVD) are the leading cause of death in Europe and worldwide. Coronary heart disease and stroke are the most common, accounting for more than 17 million deaths in 2019 [1]. A common mechanistic pathway for these diseases is the formation of atherosclerotic lesions, leading to the narrowing of arteries and potentially also exacerbating to the formation of a thrombus and hence, acute clinical events. Longitudinal studies have shown that atherosclerosis and cardiovascular risk factors appear during childhood and adolescence and tend to track into adulthood, along with lipid profile abnormalities [2], hypertension [3], and obesity [4]. Cardiovascular health is based on a combination of non-modifiable (family medical history, gender, age, ethnic background) [5] and modifiable characteristics, which can be divided into two categories—ideal health behaviors (including not smoking, body mass index (BMI) < 25 kg/m^2^, physical activity > 60 min of moderate-to-vigorous intensity per day for adolescents, and a healthy dietary pattern) and ideal health factors (including blood pressure <120/<80 mmHg, blood cholesterol < 200 mg/dL, and blood glucose < 100 mg/dL) [6]. Even though most cardiovascular risk factors are well-documented, there are still knowledge gaps about the impact of health behaviors and health factors during adolescence on cardiovascular risk factors in adulthood [7,8].

Adolescence is indeed a time of transition when lifestyle habits are changing but thereafter will persist into adult life. The Healthy Lifestyle in Europe by Nutrition in Adolescence (HELENA) study was a multicenter study performed in nine European countries (10 study centers) that included 3528 adolescents aged from 12.5 to 17.5 years, between 2006 and 2007 [9]. The main aim of the HELENA study was to collect reliable data on nutrition, health-related parameters, and a large battery of cardiovascular biomarkers [10,11], to obtain a reliable overview of the cardiovascular status of European adolescents. BELINDA (BEtter LIfe by Nutrition During Adulthood) is the longitudinal follow-up study of the HELENA adolescents, conceived for the purpose of gaining a better understanding of how cardiovascular risk evolves in the transition from a European adolescent to an adult population. Indeed, this study will allow the identification of health behaviors occurring during adolescence and the detection of subclinical atherosclerosis in young adults.

The objective of the current article is to describe the aims, design, and methods used for conducting the BELINDA study. The secondary objective is to compare the characteristics of the BELINDA study population (gender, age, weight, height, BMI, and maternal educational level) with the HELENA population not participating in the follow-up study.

## 2. Materials and Methods

### 2.1. Study Design

The BELINDA study, conducted in 2016–2020, is a follow-up of the adolescents included in four of the 10 centers from the HELENA study. The primary objective of the BELINDA study is to evaluate the association of adolescents’ health-related factors (12.5–17.5 years) and the cardiovascular risk in young adulthood (21–32 years), using the Pathobiological Determinants of Atherosclerosis in Youth (PDAY) risk score [12], which is the principal outcome measure. The secondary objective of the BELINDA study is to track the longitudinal evolution of relevant parameters from adolescence to adulthood over 10–14 years, such as dietary intake, physical activity and physical fitness [13], mental health, socioeconomic status, and anthropometry.

### 2.2. Study Population

The study population was based on the 3528 HELENA study adolescents [9], distributed over 10 European cities: Athens and Heraklion (Greece), Dortmund (Germany), Ghent (Belgium), Lille (France), Pecs (Hungary), Rome (Italy), Stockholm (Sweden), Vienna (Austria), and Zaragoza (Spain). Details of specific sampling methods have been reported elsewhere [10].

For inclusion in the BELINDA study, all the HELENA subjects in the four participating centers—Lille (France), Ghent (Belgium), Rome (Italy), and Zaragoza (Spain)—were contacted between 2016 and 2020 by mail, phone, letter by post, or social networks. There were no specific inclusion criteria, except that subjects must have participated in the HELENA study, signed an informed consent form, and had social insurance at the time of BELINDA enrollment. Excluded participants were those deprived of liberty or those not able to receive information about the study. Pregnant women had to wait at least 6 months after giving birth to participate.

### 2.3. Data Collection

BELINDA participants were invited to a one-day visit that took place in a medical setting (clinical investigation center or clinical study site). Several tests and measurements at young adult age (BELINDA) were conducted that were identical to those conducted during adolescence (HELENA—baseline measurements). However, several measurements were adapted for an adult population and a few measurements were only conducted in the BELINDA study (Table 1).

The BELINDA visits included anthropometric measurements, lifestyle habits’ evaluations (diet, physical activity, and fitness), clinical examination, blood sampling (Table 2), and biobanking (Table 3). Fresh blood, frozen blood, and stool samples were collected for measuring biomarkers, evaluating metabolic, nutritional, and inflammatory status. Nutrient deficiency, epigenetic, and psychologic biomarkers (measured by cortisol concentration in the hair sample) were also measured for secondary analysis. Parental cardiovascular history such as coronary heart disease, stroke, hypertension, type 2 diabetes, and obesity was collected as heart disease risk factors in the offspring [14]. The incentive for the subjects’ participation was the assessment of medical and nutritional status, and physical fitness check-up. The data were collected in an electronic case report file, in accordance with Good Clinical Practice. No major risks were mentioned. No serious adverse events occurred during the study and the exclusion period (7 days).

### 2.4. Data Collection and Scoring Common to HELENA and BELINDA Studies

#### 2.4.1. Clinical-Anthropometric Measures

Weight was measured in subjects wearing shorts and a T-shirt without shoes, with an electronic scale (SECA 871; SECA, Hamburg, Germany) to the nearest 0.1 kg, and height was measured with a stadiometer (SECA 225; SECA, Hamburg, Germany) to the nearest 0.1 cm. A set of skinfold thicknesses (biceps, triceps, subscapular, and suprailiac) and circumferences (relaxed and mid-flexed upper arm, waist, hip, and upper thigh) were measured on three consecutive occasions on the left side of the body, with a Holtain caliper (to the nearest 0.2 mm), and with a non-elastic tape to the nearest 0.1 cm, respectively. All anthropometric measurements were collected by the standard methodology used in the HELENA study [15], according to Lohman’s anthropometric standardization reference manual [16]. BMI (kg/m^2^) and waist-to-hips ratio were calculated. A bioelectrical impedance analysis (BIA) was also performed for body composition (BIA 101 AKERN, Akern Srl., Firenze, Italy). The blood pressure measurement was performed using an automatic blood pressure device (OMRON M6, HEM 70001; Omron, Kyoto, Japan) on at-rest patients who were seated for at least 5 min with arms supported. The measurement was taken and the arm with the highest systolic blood pressure value was chosen. The measurement was repeated every minute for 4 min, and the mean of the last two measurements was calculated and recorded in the patient’s file in the electronic case report file. Measurements were performed by the same certified person to avoid inter-personal biases.

#### 2.4.2. Dietary Intake Assessment

The HELENA-DIAT software was used in the context of the HELENA and BELINDA studies to record three different 24 h dietary recall self-reports (2 days during the week and 1 day at the weekend) and to collect information about the food habits and the quantities eaten [17]. Participants were asked to select the foods or meals they had eaten. Pictures of these dishes, dosed differently, appeared and they had to select the plate corresponding to the quantity they had eaten. Meals were converted into energy and macro/micronutrient intake primarily by using the German food code and nutrient database (Bundeslebensmittelschlüssel, BLS, version II3.1). The nutrients included in the main statistical analysis for the association with the primary end point (PDAY risk score) were salt (sodium chloride total intake, present in the food and added), long-chain polyunsaturated fatty acid (LC-PUFA), including eicosapentanoic acid (EPA) and docosahexaenoic acid (DHA) (using the German food code and nutrient database), and fructose (using fructose composition analysis [18,19]) intake, in gram per day. Data scoring was also performed to assess the Diet Quality Index (DQI), the Healthful Plant-based Diet Index (HPDI), and the NOVA classification for ultra-processed food (UPF) consumption.

The Diet Quality Index for Adolescents (DQI-A) is a semi-quantitative food frequency questionnaire used in the context of the HELENA study and was calculated using the HELENA-DIAT results. This score was validated according to the Flemish Food-Based Dietary Guideline (FBDG) and its four basic principles for a healthy and balanced diet: dietary quality, dietary diversity, dietary equilibrium, and meal patterns. For this index, the daily diet was divided into nine food groups: (1) water, (2) bread and cereals, (3) grains and potatoes, (4) vegetables, (5) fruits, (6) milk products, (7) cheese, (8) meat, fish, eggs, and substitutes, and (9) fat and oils. More details on the calculation have been reported elsewhere [20]. This same index was used for the BELINDA population but following the FBDG recommendations for adults instead of those for adolescents.

The Healthful Plant-based Diet Index (HPDI) assigned a positive score to healthy plant-based foods (whole grains, fruits, vegetables, nuts, legumes, vegetable oil, and tea or coffee) and gave a negative score to unhealthy plant-based foods (fruit juices, refined grains, potatoes, sugar-sweetened beverages, and sweets) and animal-based foods (animal fat, dairy, eggs, fish, and meat). These 18 food groups were divided into quintiles and were summed up to calculate the HPDI [21].

The NOVA classification allows food to be classified according to four levels of transformation: (1) natural, unprocessed, or minimally processed, such as fruits, vegetables, and yogurt; (2) processed culinary ingredients, which are substances derived from level 1, modified by processes that include pressing, refining, grinding, milling, and drying, such as butter or dried fruits; (3) processed foods that are level 1 substances to which level 2 substances have been added, such as cheese or canned fish; and (4) UPF, that is reconstituted, pre-prepared, or composed of food additives, such as soft drinks, low-fat products, and ready-made meals. More details about the NOVA classification have been reported elsewhere [22]. In the context of the BELINDA and the HELENA studies, the HELENA-DIAT was used to classify the food intake of the population in these four categories. The consumption was measured in kcal per day, and the UPF intake–total energy intake ratio was calculated and expressed as a percentage.

#### 2.4.3. Physical Activity

Physical activity (PA) was assessed by wearing an innovative triaxial accelerometer (ActiGraph^®^, Model GT3X; ActiGraph, Pensacola, CA, USA) for 7 consecutive days after the clinical visit [23]. This device measures sedentary time and provides information on the amount and intensity of PA by collecting data every second. Participants who did not record at least 3 days with a minimum of 10 h of activity per day were excluded from the analyses [24,25]. The measure of physical activity in young adults was divided into the four categories of (1) sedentary time, (2) light PA, (3) moderate PA, and (4) vigorous PA, and is based on the cutoff points of 0–99, 100–2000, 2001–5724, and >5724 counts/min, respectively [26,27]. In the context of the study, we used moderate-to-vigorous physical activity (MVPA, >2000 counts/min) and sedentary time (<100 counts/min).

#### 2.4.4. Physical Fitness

Cardiorespiratory fitness (CRF) was assessed using the 20 m shuttle run test to reach maximal oxygen consumption (VO2 max) [28]. Participants were required to run between two lines spaced 20 m apart, while keeping pace with pre-recorded audio signals. The initial speed was 8.5 km/h and was increased by 0.5 km/h per minute (1 min = 1 stage). Participants were instructed to run in a straight line, to pivot on completing a shuttle, and to pace themselves in accordance with the audio signals. The test was finished when the participant stopped because of fatigue or failed to reach the end lines concurrent with the audio signals twice consecutively. The test was performed once and the last completed stage or half-stage at which the subject dropped out was scored.

Upper body muscular strength (UBMS) was assessed by the handgrip test, using a hand dynamometer with an adjustable grip (Hand Grip Digital Dynamometer TKK 5401 Grip D; Takei, Japan). In the standing position, the participant squeezed gradually and continuously for at least 2 s, performing the test with their right and left hands in turn, with their elbow at full extension [29,30]. The test was performed twice and the maximum score for each hand was recorded in kilograms. The maximum score for left and right hand was used to compute the hand grip data average, which was used for the analysis. Lower body explosive strength (LBES) was assessed by the standing broad long jump test [31]. The participants were required to jump as far as possible with feet together on a non-slip hard surface from a starting position. Swinging of the arms and bending of the knees were allowed. The data recorded were the longest distance in centimeters. The evaluation was performed twice and recorded in centimeters. The maximum score was used for the analysis.

#### 2.4.5. Socioeconomic Parameters

In the HELENA study, maternal education level data were collected using a specific questionnaire [32]. Using the International Standard Classification of Education (ISCED), maternal education level, as reported by the adolescents, was categorized into four groups: primary education (ISCED level 0 or 1; score = 1); lower secondary education (ISCED level 2; score = 2); higher secondary education (ISCED level 3 or 4; score = 3); and tertiary education (ISCED level 5 or 6; score = 4). For the purposes of the study, the three lower groups were merged into one group (i.e., ‘primary education, lower secondary education, and higher secondary education’) for comparison with tertiary education (higher education or university level). In the context of the BELINDA study, data on the maternal education level were used and the education level of the subjects was collected following the same classification. 

#### 2.4.6. Dietary Intake Assessment

Fasting blood samples were collected by venipuncture between 8:00 a.m. and 10.00 a.m. to measure inflammatory biomarkers and to define nutritional (lipids, carbohydrates, vitamins, and minerals) and metabolic profiles (Table 2). The analysis was performed locally. Each local medical laboratory is accredited by their country’s competent authority, according to the certification of quality and competence ISO 15189, to perform biological and clinical measures. Processing of fresh blood samples has been detailed elsewhere [33].

### 2.5. Data Collection and Scoring Only Conducted in the BELINDA Study

#### 2.5.1. PDAY Risk Score

The primary outcome of the BELINDA study is the PDAY risk score (Table 4), derived from the PDAY study, which measured cardiovascular risk factors in 1117 subjects aged 15–34 years who died from external causes, with atherosclerotic lesions in the coronary arteries or the abdominal aorta, between 1987 and 1994 [34]. Parameters included in the score calculation are divided into two categories, non-modifiable parameters (age and gender) and modifiable parameters (HDL and non-HDL cholesterol, smoking habits, blood pressure, BMI, and glycohemoglobin (HbA1c level), which are all known as cardiovascular risk factors [13,35]. This score was calculated as described in Table 4.

#### 2.5.2. Psychological Parameters

Psychological parameters, including well-being, mental health, social status, and sleep quality were collected using questionnaires, such as the Beck Depression Index [36], the State–Trait Anxiety Inventory [37], and the World Health Organization Five Well-Being Index [38], which evaluates depression, anxiety, and well-being levels, respectively. The Pittsburg Sleep Quality Index was used to assess sleep quality, which is a risk factor for developing cardiovascular disease [39,40]. A large set of neuropsychological tests developed by the CANTAB (Cambridge Neuropsychological Test Automated Battery) research suite (Cambridge Cognition^®^; https://www.cambridgecognition.com/cantab, accessed on 15 March 2022) [41] were also performed in Ghent, Lille, and Rome. These tests assess different cognitive functions such as attention, reaction time, memory, and executive function. Inhibition of subjects was also assessed by a Stroop DKEFS (Delis–Kaplan Executive Function System) test [42].

#### 2.5.3. Environmental Determinants of Physical Activity

The ALPHA environmental questionnaire (Instruments for Assessing Level of PHysical Activity and fitness) is composed of 10 items regarding the environment and daily PA, such as the availability and safety of cycle paths, the presence of sports equipment at home, the accessibility of sports halls, the mode of transport used, and the presence of stairs at the workplace, etc. This information was collected to estimate active travel and the health environment of the subjects, and to evaluate if the environment promotes physical activity [43].

#### 2.5.4. Biobanking

A biobank of blood, hair, and stool samples was established in all centers (except for stool sampling in Zaragoza) (Table 3). One stool sample was collected to measure calprotectin as a gut inflammation biomarker [44] and the bacterial quantity and diversity of the microbiota. Indeed, the microbiota appear to be involved in the development of cardiovascular disease, obesity, and diabetes [45,46]. One hair sample was collected to measure cortisol as a chronic stress biomarker [47,48]. All biological samples were stored at –80 °C in local specialized biobank facilities until analysis, when the samples were sent to specialized laboratories, in accordance with the shipping and transport of regulated biological materials guidelines of the International Air Transport Association.

#### 2.5.5. Statistical Analysis

##### Primary Outcome Analysis and Sample Size Calculation 

For the primary objective, the association of each candidate variable at adolescence with the PDAY risk score will be assessed in a multivariable linear regression model. Hence, no formal sample size calculation was possible, but we considered the rule of thumb of 10 [49] to 20 [50] subjects per independent variable analyzed. We planned to include 330 subjects in the BELINDA study and estimated 15% missing data, corresponding to 280 subjects with all available data, permitting the inclusion of 14 independent variables (14 × 20 = 280 subjects) in the statistical analysis. 

To account for potential collinearity and to limit the instability of the final multivariable models, we will use a bootstrapping selection procedure proposed by Sauerbrei [51]. This procedure consists of performing, in each bootstrap sample (replicate), multivariable linear regression models including all candidate variables and using an automatic backward selection procedure with a removal criterion of *p*-value > 0.05. For each candidate variable and each possible pair of candidate variables (to acknowledge the collinearity issue), the proportion of replicates, in which that variable (or pair of variables) was retained in the selected model, will be determined. Variables selected in at least 70% of replicates and variables with the highest selection frequency from the pairs selected in at least 90% of replicates were retained as independent variables to build the final multivariable model.

##### Secondary Outcome Analysis

For the secondary objectives, changes in several parameters between adolescent and young adult assessments will be calculated by using linear regression models, adjusted for the baseline values of the studied parameters (i.e., assessed at adolescence). Additional adjustments will be performed on the factors identified in the primary objective.

##### Population Analysis

We compared the adolescence characteristics of the subjects participating in BELINDA with the eligible sample of the HELENA study (adolescents recruited in the four centers participating in HELENA). Quantitative variables are expressed as mean (standard deviation) or median (interquartile range), according to the normality of distribution assessed graphically and using the Shapiro–Wilk test. Categorical variables are expressed as frequencies and percentages. The magnitude of the differences was assessed by calculating the standardized differences and an absolute standardized difference of 20% was interpreted as a meaningful difference.

##### Management of Missing Data 

Missing data in candidate variables and in PDAY risk scores will be handled by simple imputation if the missing rate was <10%, or otherwise, by combining the bootstrap method with multiple imputation [52]. The imputation procedure will be conducted under the missing-at-random assumption, by using a regression switching approach (multivariate imputation by chained equations), with a predictive mean matching method for quantitative variables and logistic regression models (binomial, ordinal, or multinomial) for categorical variables [53]. The imputation procedure will be performed using all adolescence characteristics and PDAY risk scores. In the case of multiple imputation, the regression coefficients of the final multivariable model obtained in multiple imputed data sets will be combined using Rubin’s rules [54]. The goodness of fit of the final multivariable linear regression model will be assessed by reporting the R-squared and residual diagnostics (normality of residual and homoscedasticity). In the case of failure of residual diagnostics (even after applying usual transformations), a quantile regression model will be employed.

##### Statistical Analysis Software 

Data will be analyzed using SAS software (SAS Institute Inc., Cary, NC, USA) and all statistical tests will be performed with a two-tailed alpha risk of 0.05. Statistical analysis will be independently performed by the Department of Biostatistics, University of Lille.

### 2.6. Ethical Considerations and Trial Registration

The objectives were explained carefully to each participant and a written informed consent was obtained. All procedures were performed in accordance with the International Conference of Harmonization (ICH-E6) for Good Clinical Practices (GCP—R2) [55]. The BELINDA study was approved by the Ethics Committee of Lille (Comité de Protection des Personnes Nord-Ouest IV, N° CPP 16/29) for France, the Ethics Committee of Rome (Comitato etico “LAZIO 2”, Studio 04/2019) for Italy, the Ethics Committee of Ghent (Commissie voor Medische Ethiek, B670201628798, 30/05/2016) for Belgium, and the Ethics Committee of Zaragoza (Comité de Ética de la Investigación de la Comunidad de Aragón, 09/2017) for Spain. The study is registered at http://www.clinicaltrials.gov (accessed on 15 March 2022) under identifier NCT02899416.

### 2.7. Data Handling

All data were recorded by trained clinical investigators and/or by the study site coordinator using an electronic case report form (eCRF Ennov EDC^®^, Ennov 33,270 Floirac, France). Data safety and security measures were considered at every study site (restricted staff access, password protection, firewall, and virus spyware protection). To ensure the data quality, a study monitor from the trial sponsor verified and cross-checked all data against the investigator’s source document records. The essential data necessary for monitoring the primary and secondary end points were identified and were managed at regular intervals throughout the trial by the data management team of the Data Management Department of Lille University Hospital, using the predefined rules. In the case of discrepancies, queries were sent to the investigator, field researchers, and study site coordinator for resolution.

## 3. Results

### 3.1. Participation Rate and Database

Figure 1 presents the flow diagram of the study. Among the 10 centers and the 3528 subjects included in the context of the HELENA study, four centers agreed to participate in the recruitment for the BELINDA study. A total of 1327 subjects were eligible. Among these 1327 subjects, 1095 were unreachable, untraceable, refused, or were unable to participate because of the COVID-19 pandemic. The final database contains a total of 232 subjects (participation rate = 18%), assessed in the HELENA study and reassessed 10–14 years later in the context of BELINDA, versus 1095 adolescents who did not participate in BELINDA. Table 5 presents the details of the inclusions by investigation center. The six other centers were not able to conduct the study for different reasons (logistics, human resources). Recruitment was interrupted just before the outbreak of the COVID-19 pandemic (February 2020).

### 3.2. Comparison of Descriptive Data between Participants and Non-Participants

The secondary objective of the present article was to characterize the study population at adolescence compared to the HELENA population not participating in the BELINDA study. Table 6 presents data related to gender, age, weight, height, BMI, and maternal education level in these two populations. There were no significant differences between these two populations regarding gender, age, height, and weight. The HELENA population included in the BELINDA study had a significantly lower BMI (*p* < 0.001) and a significantly higher maternal education level (*p* = 0.027) compared to the HELENA population not participating in the BELINDA study.

## 4. Discussion

### 4.1. Reliability of Dietary Intake Measures

Parameters recognized to have an impact on modifiable cardiovascular risk factors were identified [19,56,57,58]. Salt, EPA + DHA, and fructose intakes will be included in the multivariate linear regression to analyze their impact on the PDAY score (Table 7).

Three nutritional indices will then be included in the statistical analysis, the DQI, the HPDI, and the NOVA classification for UPF. The DQI is based on adherence to the Flemish FBDG, which is associated with a better cardiovascular risk profile and less inflammation [59]. The HPDI is based on plant-based food consumption, which is associated with a lower risk of CVD [20]. The NOVA classification is a recognized tool based on the consumption of UPF, which is increasingly studied because of an increased risk of cancer [60] and cardiovascular diseases in the long term [61].

### 4.2. Reliability of Physical Activity, Fitness and Sedentarism Variables

Physical activity is associated with a lower risk of CVD [62]. We used accelerometry here because it is the gold standard method of objectively assessing PA. We will include the sedentary and the moderate-to-vigorous time (mean of 7 days) in the statistical analysis (Table 7). 

As physical fitness has an impact on cardiovascular health and lipid profile [63], we performed a detailed assessment using CRF, UBMS, and LBES tests, which are recognized as the most reliable tools to evaluate physical fitness [64]. 

### 4.3. PDAY Risk Score

The PDAY risk score (Table 4) is a reliable tool to estimate the probability of having advanced atherosclerotic lesions in the coronary arteries and the abdominal aorta in young adults, in the long term [12]. Indeed, all candidate variables included in the score are modifiable, objective, accurate, and were measured with the same methodology in the four centers (Table 7). It has been previously used and validated in the CARDIA study (*n* = 3008, aged from 18 to 30 years) [65] and the Cardiovascular Risk in Young Finns Study (*n* = 1279, aged from 12 to 39 years) [66].

### 4.4. Studied Population

The lower participation rate for the BELINDA study compared to the HELENA study is explained by the large time interval between the two studies. Indeed, less than 20% of the HELENA subjects from the four centers participating in BELINDA could be reassessed. Most of the eligible subjects who were not reviewed could not be reached despite our best efforts (because of change of address, telephone number, or e-mail address), and a small proportion refused to participate because of unavailability, or because they had moved too far away between adolescence and adulthood. As mentioned before, adolescence is a transition period, and during this 10-year period, family structure and employment changes can easily explain why we were not able to contact families again. In addition, recruitment was still active in 2020 in the French (Lille) and the Italian (Rome) centers and had to be interrupted due to the COVID-19 pandemic. To achieve the population target with complete data, we decided to open two more centers (Zaragoza and Rome) in addition to Lille and Ghent. However, there is a large time lapse in the recruitment period per center, which will be taken into consideration for analysis (Table 5). As mentioned in the statistics section, the number of analyzed variables was calculated depending on the study population. Between 10 and 20 subjects are needed per analyzed variable. However, increasing the number of subjects per variable would provide better statistical power. At baseline, we planned to include 280 subjects without missing data. The premature stop of the inclusions made us revise our target sample size downwards and 232 subjects were finally included. In order to have 20 subjects per variables, 11 variables were identified as candidates for the primary outcome analysis (Table 7).

The higher maternal education level in the BELINDA participants compared with the eligible participants reflects the difficulty of involving disadvantaged and low social class populations (less easily reachable, more likely to relocate, and more skeptical about participation in clinical studies in general) [67]. This difference explains the lower weight and BMI in the BELINDA population. Indeed, other studies have shown a negative association between the education level of the mother and the weight and BMI in a child and adolescent population (5 to 18 years) [68].

### 4.5. Strenghts and Limitation

The strengths of the BELINDA study are its longitudinal design, allowing for long-term follow-up and the collection of information over a period of 10–14 years in the same subjects. The diversity of the analyzed parameters also allows for a multivariate approach. Moreover, the same field researchers worked on the HELENA and BELINDA studies, guaranteeing the reproducibility and reliability of data collection. The expertise of the centers in various fields, i.e., clinical research (methodology, statistics, and data management), nutrition, physical activity, cardiovascular health, and well-being, ensures the quality of the collected data. Using the HELENA database is a unique opportunity to identify new parameters that could have an impact on cardiovascular risk.

The weakness of the study was the low participation rate due to the difficulty of contacting participants 10–14 years later, the HELENA adolescents as young adults which yielded a low sample size, selection, and recall bias in the BELINDA population. The failure to reach the target of 280 participants due to COVID-19 resulted in a decrease in the variables included in the primary endpoint multivariable model, from 14 to 11, in order to have at least 20 subjects per variable, as mentioned in the statistics section.

## 5. Conclusions

The BELINDA study is a longitudinal study from adolescence to adulthood (i.e., a period of 10–14 years for the same subjects), allowing some health and lifestyle data collection in both periods in order to highlight cardiovascular risk profiles and their prediction by the evolution of dietary, physical activity, and fitness habits in the transition from a European adolescent to adult population. The recruitment was difficult due to recall bias and the COVID-19 pandemic. In this context, HELENA subjects who were participants or non-participants in the BELINDA study were similar, except for BMI and maternal educational level, showing a selection bias that will need to be taken into consideration for future analyses. This article will thus serve as a methodological basis for future analysis of the BELINDA population, with the ultimate goal of identifying factors for the prevention of cardiovascular diseases.

## Figures and Tables

**Figure 1 nutrients-14-02089-f001:**
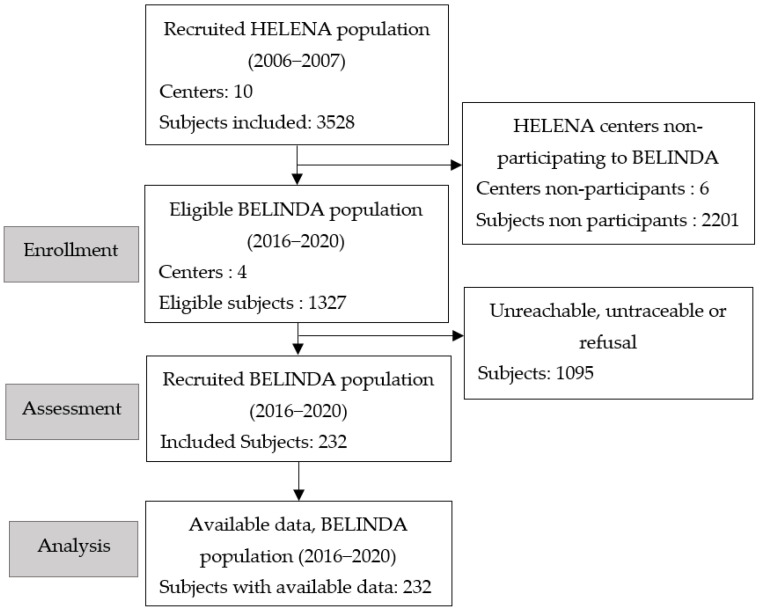
Flow diagram of the BELINDA study recruitment process. Shadow corresponds to the inclusion steps.

**Table 1 nutrients-14-02089-t001:** Data set, tools, and process shared by HELENA and BELINDA studies.

Parameters	HELENA	BELINDA
Anthropometrics and vital signs measures	X	X
Nutritional data collection and coding Process *	X	X
Physical activity level	X	X
Physical fitness level	X	X
Socioeconomic status	X	X
Biological analysis (fresh blood sample) **	X	X
PDAY risk score		X
Psychological tests		X
Environmental data		X
Biobanking (blood, stool, and hair samples)		X

* Including Diet Quality Index, NOVA ultra-processed food, and Healthful Plant-based Diet Index. ** Including plasma inflammatory biomarkers, lipid profile, and vitamin and mineral concentration.

**Table 2 nutrients-14-02089-t002:** Blood sampling process and parameters analyzed in the BELINDA study.

Type of Blood Sample	Parameters	Objective of the Analysis
EDTA tube	HbA1c White blood cells count	Glycemic profile, **PDAY risk score** Nutrient deficiency
Lithium–Heparin tube	Albumin ALT AST γ-GT Calcium, phosphorus, proteins Glucose	Nutrient deficiency Liver health Nutritional profile Glycemic control
Serum tube	HDL cholesterol Total cholesterol Triglycerids 25-OH cholecalciferol VitD Ferritin Insulin, leptin CRP	Lipid profile, **PDAY risk score** Lipid profile, **PDAY risk score** Lipid profile Nutritional profile Nutritional profile Metabolic profile Inflammation

Abbreviations: HbA1c: glycohemoglobin; ALT: alanine aminotransferase; AST: aspartate aminotransferase; γ-GT: gamma glutamiltranspeptidase; CRP: C-reactive protein; HDL: high-density lipoprotein; EDTA: ethylenediamine tetraacetic acid; PDAY: Pathobiological Determinants for Atherosclerosis in Youth (writing in bold as the primary outcome).

**Table 3 nutrients-14-02089-t003:** Blood, stool, and hair samples, preanalytic procedure, and parameters for BELINDA biobanking.

Type of Biological Samples	Preanalytic Procedures	Analysis	Objective of the Analysis
Blood sample EDTA 9 mL (−80 °C)	Centrifugation (Red blood cells)	PUFA, HbA1c	Lipid profile Glycemic profile, **PDAY risk score**
	Centrifugation (Buffy coat)	DNA methylation Telomere length	Epigenetics Cellular senescence/aging
	Centrifugation (Plasma)	CML	Inflammation
Blood sample serum tube 7 mL (−80 °C)	Centrifugation (Serum)	ICAM-1, IL-10, IL-6, TNF-α, VCAM TMAO	Inflammation Metabolomics
Stool sample (−80 °C)	No technical procedure	Calprotectin, microbiota, SCFA	Intestinal health Lipid profile
Hair sample (room temperature)	No technical procedure	Cortisol	Stress

Abbreviations: EDTA: ethylenediamine tetraacetic acid; PUFA: polyunsaturated fatty acid; HbA1c: glycohemoglobin; CML: carboxymethyllysine; ICAM: intracellular adhesion molecule; IL: interleukin; TNF: tumor necrosis factor; VCAM: vascular cell adhesion molecule; SCFA: short-chain fatty acid; PDAY: Pathobiological Determinants for Atherosclerosis in Youth (writing in bold as the primary outcome); TMAO: trimethyulamine N-Oxide.

**Table 4 nutrients-14-02089-t004:** Pathobiological Determinants for Atherosclerosis in Youth (PDAY) risk score, predicting target lesions in the coronary arteries and the abdominal aorta.

	PDAY Risk Score Point Value	
Risk Factors	Coronary Arteries	Abdominal Aorta
Age (years)		
15–19 *	0	0
20–24	5	5
25–29	10	10
30–34	15	15
Sex		
Male *	0	0
Female	−1	1
Non-HDL cholesterol (mg/dL)		
<130 *	0	0
130–159	2	1
160–189	4	2
190–219	6	3
≥220	8	4
HDL cholesterol (mg/dL)		
<40	1	0
40–59 *	0	0
≥60	−1	0
Tobacco consumption		
Nonsmoker *	0	0
Smoker	1	4
Blood pressure		
Normotensive * (MAP < 110 mmHg)	0	0
Hypertensive (MAP ≥ 110 mmHg)	4	3
Obesity (BMI ≥ 30 kg/m^2^)		
Male		
No *	0	0
Yes	6	0
Female		
No *	0	0
Yes	0	0
Hyperglycemia (% HbA1c)		
<8 *	0	0
≥8	5	3

Abbreviations: HDL: high-density lipoprotein; BMI: body mass index; MAP: mean arterial pressure; HbA1c: glycohemoglobin; * Reference category.

**Table 5 nutrients-14-02089-t005:** BELINDA population detailed by investigation center.

Investigation Center	HELENA Population (*n* = 1327)	BELINDA Population (*n* = 232)	BELINDA Recruitment Period
Ghent (Belgium)	336	86	November 2016—February 2017 (4 months)
Lille (France)	287	72	February 2017–February 2020 (3 years)
Rome (Italy)	304	19	July 2019–February 2020 * (8 months)
Zaragoza (Spain)	384	55	November 2017–June 2018 (8 months)

* Since 2021, the Italian center has restarted subject recruitment.

**Table 6 nutrients-14-02089-t006:** Characteristics at adolescence of included and non-included subjects.

	Not Participating	Participating	*p*-Value	Absolute Standardized Difference
Number of adolescents (*n*)	1095	232		
Gender (*n* (%boys))	493 (44.7)	107 (45.9)	0.72	−0.0255
Age (years ± SD)	14.8 ± 1.2	14.9 ± 1.2	0.60	−0.0377
Height (cm ± SD)	165.0 ± 9.2	166.1 ± 9.7	0.11	0.1149
Body mass (kg ± SD)	57.9 ± 12.1	56.4 ± 10.9	0.083	−0.1299
BMI (kg/m^2^ ± SD)	21.2 ± 3.6	20.4 ± 3.1	**<0.001**	−0.2370
Maternal education level ½ * (%)	61.4/38.6	53.2/46.8	**0.027**	0.1447

* Maternal education level 1: primary/secondary school level; level 2: high school or university level. Bold writing corresponds to the statistically significant results.

**Table 7 nutrients-14-02089-t007:** List of the HELENA adolescent characteristics included in BELINDA primary outcome analysis, according to the study sample size calculation.

Type of Variable		Parameters Analyzed
**Nutrition**	1	Salt intake
	2	EPA + DHA intake
	3	Fructose intake
	4	Diet Quality Index
	5	Healthful Plant-based Diet Index
	6	NOVA ultra-processed food score
**Physical activity and fitness**	7	Physical activity (MVPA)
	8	Sedentary time
	9	CRF
	10	UBMS by hand grip
	11	LBES by standing broad jump

Abbreviations: EPA: eicosapentaenoïc acid; DHA: docosahexaenoic acid; MVPA: moderate-to-vigorous physical activity; CRF: cardiorespiratory fitness; UBMS: upper body muscular strength; LBES: lower body explosive strength.

## Data Availability

Not applicable.

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
