# Peer review of "Identification of Lifestyle Risk Factors in Adolescence Influencing Cardiovascular Health in Young Adults: The BELINDA Study"

_nutrients, 2022, doi:10.3390/nu14102089_

Round 1

Reviewer 1 Report

Morcel et al. aimed to identify lifestyle risk factors in adolescence influencing cardiovascular health in young adults.

It is unclear which method was used to assess salt intake.

I wonder how the authors chose the biomarkers that will be used in the study (table 2 and 3).

The study has no major methodological flaws, but perhaps my biggest question is what is in fact the point of the present study? For instance, what is the use of comparison between HELENA and BELINDA, since the compared variables don't seem to be quite useful. On the other hand, final results of the presented trial will be interesting, but at this point they provide scarce amount of information for the reader.

Author Response

Dear reviewcer, please see the attachment.

Thank you

Jules MORCEL

Reviewer 2 Report

This is an original article about a longitudinal study from adolescence to adulthood, with the aim to highlight cardiovascular risk profiles and their prediction by the evolution of dietary, physical activity, and fitness habits in the transition from European adolescent toward an adult population. The manuscript is well written and well structured. It deals with an interesting topic, where further studies are needed. Tables are clear and useful for summarizing the results.

Author Response

Dear reviewer, please see the attachment.

Thank you

Jules MORCEL
